# Activation of Esterase D by FPD5 Inhibits Growth of A549 Lung Cancer Cells via JAB1/p53 Pathway

**DOI:** 10.3390/genes13050786

**Published:** 2022-04-28

**Authors:** Wen Yao, Yuejun Yang, Xinpeng Chen, Xiaoling Cui, Bangzhao Zhou, Baoxiang Zhao, Zhaomin Lin, Junying Miao

**Affiliations:** 1Shandong Provincial Key Laboratory of Animal Cells and Developmental Biology, School of Life Science, Shandong University, Qingdao 266237, China; 201912314@mail.sdu.edu.cn (W.Y.); 202020325@mail.sdu.edu.cn (Y.Y.); kyleshandong@163.com (X.C.); c1490016642@163.com (X.C.); zbz990716@163.com (B.Z.); 2Hubei Key Laboratory of Edible Wild Plants Conservation & Utilization, National Demonstration Center for Experimental Biology Education, School of Life Science, Hubei Normal University, Huangshi 435002, China; 3Institute of Organic Chemistry, School of Chemistry and Chemical Engineering, Shandong University, Jinan 250100, China; bxzhao@sdu.edu.cn; 4Institute of Medical Science, The Second Hospital of Shandong University, Jinan 250033, China

**Keywords:** Esterase D, FPD5, p53, JAB1, cell cycle, cell growth

## Abstract

Esterase D (ESD) is widely distributed in mammals, and it plays an important role in drug metabolism, detoxification, and biomarkers and is closely related to the development of tumors. In our previous work, we found that a chemical small-molecule fluorescent pyrazoline derivative, FPD5, an ESD activator, could inhibit tumor growth by activating ESD, but its molecular mechanism is still unclear. Here, by using RNA interference (RNAi), andco-immunoprecipitation techniques, we found that ESD suppressed the nucleus exportation of p53 through reducing the interaction between p53 and JAB1. The protein level of p53 in the nucleus was upregulated and the downstream targets of p53 were found by Human Gene Expression Array. p53 inhibited the expression of *CDCA8* and *CDC20*. Lastly, the cell cycle of A549 cells was arrested at the G0/G1 phase. Together, our data suggest that ESD inhibited the cancer cell growth by arresting the cell cycle of A549 cells via the JAB1/p53 signaling pathway. Our findings provide a new insight into how to inhibit the growth of lung cancer with the activation of ESD by FPD5.

## 1. Introduction

Esterase D (ESD) is a non-specific esterase widely distributed in various organisms and also named S-Formylglutathione Hydrolase (SFGH) [1,2]. It is a member of the carboxylesterase family and has both carboxylesterase and thioesterase activities. The gene of ESD, located on the long arm of human chromosome 13, is usually used as a genetic diagnostic marker [3,4]. Meanwhile, ESD plays an important role in the process of glutathione-dependent detoxification, regulating cholesterol efflux and virus infection in humans, and is closely related to the development of tumors [5,6,7,8]. Lower ESD activity in lung adenocarcinoma patients represents a higher tumor grade [9]. Considering the advantages of the small-molecule chemical, we recently found that FPD5, 4-chloro-2-(5-phenyl-1-(pyridin-2-yl)-4,5-dihydro-1H-pyrazol-3-yl) phenol, could activate ESD and downregulate the viability of cells including A549, HeLa, and H322 cells [10,11]. FPD5 will be a good tool for the further study of ESD function. However, the molecular mechanism is not clear. This research aims to answer this question.

p53 is a tumor suppressor and maintains low levels in normal cells [12]. p53 is significant for regulating cell proliferation, the cell cycle, metastasis, apoptosis, senescence, and autophagy [13,14]. When cells are exposed to various forms of stress, p53 tightly regulates cell growth through inducing DNA repair, cell senescence, cell cycle arrest, or apoptosis [15,16,17]. However, the function of p53 depends on its cellular localization [18]. For example, p53 located in the nucleus can promote autophagy by regulating the transcription of the mTOR signaling pathway and autophagy-related gene ATGs in a transcription-dependent manner, but cytoplasmic p53 inhibits autophagy [19,20,21]. JAB1 can promote the nuclear output of p53 and assist MDM2, an E3 ubiquitin ligase, to promote the degradation of p53 in the cytoplasm [22]. In our previous studies, we have proven that activating ESD by FPD5 deacetylates threonine 89 of JAB1. We demonstrate here that ESD increases the nuclear localization of p53 by JAB1, and it is vitally important to find a new factor to regulate p53 and inhibit tumor growth.

Here, we found that a novel factor esterase D (ESD) reduced the interaction between JAB1 and p53 to suppress cancer cell growth. We reveal that ESD increased the protein level of nucleus p53 by inhibiting the interaction between JAB1 and p53 and decreasing the nuclear exportation of p53. Thereby, p53 inhibited the expression of *CDCA8* and *CDC20*, and arrested the cell cycle of A549 cells at the G0/G1 phase. Our research opens up a new approach for future studies of the ESD-mediated signaling pathway.

## 2. Materials and Methods

### 2.1. Antibodies and Materials

Antibodies for p53 (10442-1-AP) were purchased from Proteintech (Chicago, IL, USA). ESD (sc-134333), goat anti-mouse horseradish peroxidase (HRP)-conjugated IgG (GP016129) was purchased from Santa Cruz Biotechnology (Santa Cruz, CA, USA). JAB1 (ab124720) was purchased from Abcam. GAPDH (G8795) and β-actin (ACTB, A1978) were purchased from Sigma-Aldrich (Burlington, MA, USA). Lamin A/C (4C11) and Lamin B1 (D9V6H) were from Cell Signaling Technology (Danvers, MA, USA). Propidium iodide solution (PI, P4864) was from Sigma-Aldrich (Burlington, MA, USA). Recombinant RNase A (B600476-0200) was purchased from Sangon Biotech (Shanghai, China).

The small-molecule chemical FPD5 was provided by Professor Baoxiang Zhao (Shandong University, Jinan). The synthetic protocols have been published.

### 2.2. Cell Culture

A549 cells (purchased from the ATCC (American Type Culture Collection, Manassas, VA, USA)) were cultured in RPMI medium 1640 (Gibco, 3180-022, New York, NY, USA) containing 10% calf serum (CS, *v*/*v*) (Biological Industries, 04-102-1A, Kibbutz Beit Haemek, Israel), 50 U/mL penicillin, and 50 μg/mL streptomycin (Invitrogen, 10378-016, Waltham, MA, USA) at 37 °C in a saturated humidified incubator with 5% CO_2_.

### 2.3. Western Blot Analysis

We plated cells on cubbies to 70% density. After treatments, the cells were washed thrice with pre-cooled PBS and lysed in cell lysis buffer for western and IP (Beyotime Biotechnology, P0013, Beijing, China) containing 1 mM PMSF. After centrifuging for 15 min at 12,000 rpm, 4 °C, supernatants were collected and the concentration was measured by BCA assay. Then, 30 μg/lane protein samples were loaded onto a 12% SDS–polyacrylamide gel, and transferred to PVDF membranes (Millipore, IPVH00010, Hong Kong, China). The PVDF membranes were incubated with primary antibodies for 16 h at 4 °C, and then incubated with secondary antibodies that linked horseradish peroxidase and detected by an enhanced chemiluminescence detection kit (Thermo Fisher, 34,080, Waltham, MA, USA). Infrared secondary antibodies were imaged. The relative quantity of proteins was analyzed by ImageJ 1.44P software (National Institutes of Health, Bethesda, MD, USA) and normalized to loading controls.

### 2.4. Co-Immunoprecipitation (Co-IP)

For co-immunoprecipitation, lysates (0.5 mg of protein) of cells were incubated with primary antibody (2.0 μg) for 24 h at 4 °C, with 20 μL of protein A/G Sepharose beads (Beyotime, Beijing, China). The immunoprecipitated complexes attached to agarose beads were washed three times with 400 μL PBS (containing 1 mM PMSF) and then isolated by centrifugation at 4 °C, 2000 rpm for 5 min. Then, the immunoprecipitates were eluted with IP lysate and boiled with 25 μL 2 × SDS loading buffer at 100 °C for Western blot.

### 2.5. Nuclear and Cytoplasmic Extraction

The nuclear and cytoplasmic extraction was performed using an NE-PER^TM^ Nuclear and Cytoplasmic Extraction Reagents kit (ThermoFisher Scientific, 78835, Waltham, MA, USA). A549 cells were treated with FPD5 for 24 h and were washed thrice with 1 × PBS. The cells were digested from the dish with 0.25% trypsin and collected in a 1.5 mL centrifuge tube. Cells were centrifuged at 500× *g* for 5 min. Then, we added 200 μL of CER1 to the cell pellet and incubated the suspension on ice for 15 min. We added 11 μL CER II, vortexed for 10 s, incubated the tube on ice for 90 s, and centrifuged at 16,000× *g* for 10 min. We transferred the supernatant (cytoplasm extraction) to a new centrifuge tube. We resuspended the precipitation (nucleus) with 50 μL NER after washing twice with pre-cooled PBS. We vortexed it for 20 s and incubated it for 20 min on ice, and then centrifuged it for 10 min at 4 °C, 16, 000× *g*. The resulting supernatant was subjected to nuclear extraction and was used for subsequent Western blot analysis.

### 2.6. Cell Staining for Immunofluorescence Microscopy

Cells were fixed for 20 min with 4% paraformaldehyde at room temperature after being treated with FPD5 and blocked with normal goat serum (1:10). Then, cells were incubated with primary antibodies overnight at 4 °C, and washed with phosphate-buffered saline three times and incubated with secondary antibody (1:200) for 55 min at 37 °C in the dark. LSM900 (Zeiss, Oberkochen, Germany) was used to capture the fluorescence at the indicated excitation wavelength. The fluorescence intensity was analyzed using ZEN 2010 software (Carl Zeiss, Oberkochen, Germany) for at least 10 regions of each labeling condition, and representative results are shown.

### 2.7. RNA Interference (RNAi)

To knock down the expression of ESD or JAB1, RNA interference was performed as follows. The A549 cells were transfected with ESD or JAB1 siRNA using Lipofectamine 2000 (Invitrogen, 1803709, Waltham, MA, USA) for 24 h according to instructions. The efficiency of gene knockdown was evaluated by Western blot. The siRNA ESD was synthesized and purchased from GenePharma (Shanghai, China) and siRNA JAB1 was synthesized and purchased from Tsingke (Beijing, China). SiRNA ESD sequence (5′–3′): GCUACCCACCUUGUGAAAUTT, AUUUCACAAGGUGGGUAGCTT; SiRNA JAB1 sequence (5′–3′): UGUUCUUGUUGGAUCAAUCTT, GAUUGAUCCAACAAGAACATT.

### 2.8. Quantitative Real-Time PCR (qPCR)

The TRIzol reagent method (Invitrogen, 15596018, Waltham, MA, USA) was used to extract total RNA from A549 cells. The quality of RNA was assessed by Nanodrop 2000 equipment (ThermoFisher Scientific, Waltham, MA, USA). The reverse transcription step involved the use of oligo (dT) primers, and then quantitative real-time PCR (Roche, Light Cycler 2.0 system, Basel, Switzerland) was performed. The primer pair sequences for CDCA8 were forward, 5′-CCGTGAAGTGGAAATACGAATC-3′, and reverse, 5′-GGATCTCGATGTTGTAGAGGTT-3′; CDC20, forward 5′-GGAGTGCAAGCTCTGGTGACATC-3′ and reverse, 5′-GGTGCCCACAGCCAAGTAGTTG-3′, and β-actin, forward, 5′-ACCACAGTCCATGCCATCAC-3′ and reverse, 5′-TCCACCACCC TGTTGCTGTA-3′, as a housekeeping gene. The relative level of gene expression was normalized to the β-actin level. The QuantiTect SYBR Green PCR kit (QIAgen, 204143, Dusseldorf, Germany) and LightCycler 2.0 system (Roche, Switzerland, Basel) were used in the quantitative PCR (qPCR) reactions. The reactions were performed in a 20 μL volume with 10 μL of 2  ×  SYBR Green PCR MasterMix, 0.4 μL forward primer and 0.4 μL reverse primer, 7.2 μL RNase free ddH_2_O, and 2 μL cDNA. The reaction was cycled 40 times, including pre-denaturation at 95 °C for 5 min, denaturation at 95 °C for 30 s, annealing at 58 °C for 30 s, and extension at 72 °C for 30 s. The fold change of RNA level was calculated by the 2^−ΔΔCt^ method with MxPro 4.00 (Stratagene, San Diego, CA, USA).

### 2.9. Flow Cytometric Analysis of Cell Cycle Distribution

After treatment with 5 μM FPD5 for 24 h, the A549 cells were harvested and fixed with 70% ethanol, and then incubated with 200 μL PBS containing 20 μg/mL RNase A (Sangon Biotech, B600476, Shanghai, China) for 30 min at 37 °C and stained with 50 μg/mL propidium iodide (PI, Sigma-Aldrich, P4864, Burlington, MA, USA) containing 20 μg/mL RNase A at room temperature for 30 min. The stained cells were analyzed using an Imaging Flow Cytometer (Merck, Darmstadt, Germany). The cell cycle distribution was analyzed using IDEAS software 6.0 (Merck, Darmstadt, Germany).

### 2.10. Quantitative Analysis and Statistical Analysis

The software ImageJ 1.44P (National Institutes of Health, Bethesda, MD, USA) was used to analyze the quantification of the Western blot. All confocal images were obtained from a laser scanning confocal microscope, the LSM900 (Zeiss, Oberkochen, Germany). The statistical data were from at least three independent tests. Data were expressed as mean ± SEM and were analyzed by SPSS 11.5 (SPSS Inc., Chicago, IL, USA). Data were analyzed by one-way ANOVA (followed by the Scheffé F test for post-hoc analysis). *p* < 0.05 was considered statistically significant.

## 3. Results

### 3.1. FPD5 Increased the Protein Level of p53 in the Nucleus of A549 Lung Cancer Cells

In our previous research, we found that JAB1, an ESD-interacting protein, was acetylated on the 89th threonine residue. After being activated by FPD5, ESD promoted its interaction with JAB1 and acetylated JAB1 at Thr89, resulting in an enhanced level of phosphorylation at this site competitively [23]. JAB1 had been shown to be a promoter of p53 nuclear exportation and cytoplasmic degradation, contributing to the maintenance of low intracellular p53 protein levels under normal conditions [22]. p53, one of the most important tumor inhibitors in the human body, has a good inhibitory effect on tumor occurrence, growth, and migration. Previously, we found that the chemical small molecule FPD5 inhibited tumor growth and cell viability significantly [24], and we speculated that this effect was related to p53.

Firstly, we treated A549 cells with 0, 0.1, 1, 5, 10, 20 μM FPD5 for 24 h and 5 μM FPD5 for different times. We tested the protein level of p53 through Western blot analysis. It was found that FPD5 significantly increased the overall protein level of p53 in the cells (Figure 1A–D), indicating that FPD5 could regulate the protein level of p53.

The tumor suppressor effect of p53 depends on its nuclear localization, and its DNA-binding domain can directly bind to the promoters of all kinds of genes and regulate the transcription of them [13]. Next, we tested whether ESD had an effect on the intracellular localization of p53 in the cells. A549 cells were treated with different concentrations of FPD5 for 24 h, and the nucleus and cytoplasm were separated for Western blot detection. The results showed that FPD5 significantly increased the protein levels of p53 in the nucleus, without affecting the level in the cytoplasm (Figure 1E–G). These results indicated that the activation of FPD5 could increase the level of p53 in the nucleus and increase the overall protein level in the cells, which is beneficial to the transcriptional regulation of p53 in the nucleus.

### 3.2. The Upregulation of p53 by FPD5 Was Associated with ESD

FPD5 is an activator of ESD, which has been proven before; in order to confirm that the regulation of the intracellular p53 protein level by FPD5 is related to ESD, we used ESD siRNA to knock down the expression of intracellular ESD. Compared with the cells transfected with scramble RNA, the protein level of p53 could not be upregulated by FPD5 in A549 cells transfected with ESD siRNA (Figure 2A–C).

Next, in order to determine the mechanism of ESD regulating the subcellular localization of p53, we detected whether there was a direct interaction between ESD and p53. We used p53 antibody for co-immunoprecipitation in A549 cells treated with 5 μM FPD5. Interestingly, we did not detect the ESD signal in the end (Figure 2D), indicating that there was no obvious interaction between ESD and p53. The result was also confirmed in immunofluorescence experiments (Figure 2E). In general, the activation of ESD by FPD5 increased the protein level of p53 in A549 cells, but ESD did not play a direct role in this process.

### 3.3. ESD Regulated p53 via JAB1 and Inhibited the Interaction between JAB1 and p53

In addition, the activation of ESD by FPD5 promoted its interaction with JAB1, a promoter of the nuclear exportation and subsequent cytoplasmic degradation of multiple proteins, including p53, so we speculated that ESD regulated p53 via JAB1. Using RNA interference technology to interfere with the gene expression of JAB1 in A549 cells, we firstly screened the siRNA concentration with the best interference effect at 60 nM (Figure 3A–C). Then, the protein level of p53 was detected in the A549 cells treated with 5 and 10 μM FPD5 for 24 h after being transfected with 60 nM JAB1 siRNA for 24 h. Surprisingly, FPD5 could no longer increase the overall protein level of p53 after knocking down the expression of JAB1 (Figure 3D,E), indicating that ESD regulated p53 via JAB1.

The interaction between ESD and JAB1 led to the modification of JAB1 at Thr89, so we hypothesized that this modification might be detrimental to the interaction between JAB1 and p53, thereby reducing the nuclear exportation and cytoplasmic degradation of p53. We performed co-immunoprecipitation in A549 cells with antibodies against p53 and found that FPD5 reduced the interaction of JAB1 with p53 (Figure 3F,G), which confirmed our hypothesis.

### 3.4. Activation of ESD Inhibited the Cell Cycle Progress of A549 Cells

Activating ESD inhibited the cell viability in vitro and tumor growth in vivo, but did not promote the occurrence of apoptosis [24]. p53 inhibited the tumor growth mainly by inhibiting the cell cycle progress and promoting apoptosis. We speculated that the tumor growth inhibitory effect of ESD might be related to the inhibition of the cell cycle. After treatment with 5 μM FPD5 for 24 h, A549 cells were stained with propidium iodide (PI) for flow cytometry analysis. We found that FPD5 increased the proportion of cells in the G0/G1 phase from 68.31% to 86.26% compared with the control (Figure 4A–D). This indicated that the activation of ESD significantly inhibited cell cycle progression and arrested the cell cycle of A549 cells at the G0/G1 phase. This illustrated that activating ESD deflected the cell cycle via p53.

### 3.5. p53 Suppressed the Gene Expression of CDCA8 and CDC20

In order to further search for the expression of target genes regulated by ESD through p53, we treated the A549 cell line with 5 μM FPD5 for the PrimeView Human Gene Expression Array.

We performed KEGG pathway enrichment analysis and GO enrichment classification on the genes whose expression was altered (Figure 5A), and found that these genes were related to growth as were framed with red in Figure 5C and the p53 signaling pathway as were framed with red in Figure 5B.

Among these genes, we succeeded in finding genes regulated by p53, and the genes that were downregulated more than 1.50-fold are listed in Table 1.

Among the genes listed in Table 1, *CDCA8* and *CDC20* were usually highly expressed in tumor cells and were associated with the promotion of cell cycle progression and tumor growth. Their gene expression was inhibited by p53. After treatment with different concentrations of FPD5 for 24 h, we verified the mRNA levels of these three genes in A549 cells by qPCR technology, and found that FPD5 significantly reduced the levels of the mRNA of *CDCA8* and *CDC20* compared with the control group (Figure 6A,B).

## 4. Discussion

ESD is a non-specific esterase widely distributed in mammalian cells. Recently, more and more studies have focused on the relationship between ESD activity and multiple pathological processes, such as retinoblastoma, lung adenocarcinoma, viral immunity, etc. [7,25,26,27,28,29,30,31]. However, there are few studies on ESD in tumor suppression. The direct regulation of specific target proteins and signaling pathways by small-molecule drugs is widely used in cancer therapy [32]. In this study, we found for the first time that ESD inhibited the cell cycle progress of A549 cells through the JAB1/p53 signaling pathway and had a significant inhibitory effect on tumor growth. This will not only provide new therapeutic targets and schemes for the clinical treatment of lung cancer, but also provide a theoretical basis for the development of new drugs to inhibit tumor growth. p53 regulates the expression of genes involved in regulating cell proliferation and growth in the nucleus [33]. There is increasing evidence that mutated p53 plays an important role in tumorigenesis, invasion, and migration [34]. JAB1, also named as the fifth subunit of COP9 signalosome (CSN5 or COPS5), interacts with Hdm2 to promote p53 nuclear exportation. Then, MDM2 mediated p53 ubiquitination and degradation in the cytoplasm [35,36]. In this study, we found that activating ESD by FPD5 significantly inhibited the interaction between JAB1 and p53, which would reduce the effect of nuclear exportation of p53 by JAB1, which increased the nuclear localization of p53 and provided a good condition for p53 to regulate other genes’ expression in the nucleus. However, we speculated that there were two possible reasons that activated ESD inhibited the interaction between and p53. The first possibility is that JAB1 can no longer bind to p53 after interacting with ESD. The second possibility is that the acetylation modification of JAB at Thr89 is very important for its binding to p53. The acetylation modification of JAB1 at Thr89 was removed by ESD after activation by FPD5, resulting in a reduction in its binding ability to p53. We will research the effect of this site on the binding ability of JAB1 to p53 through the threonine point mutant T89A for further verification.

In summary, our study discovered for the first time the molecular mechanism of ESD inhibiting tumor growth through the JAB1/p53 signaling pathway, which will provide a good theoretical basis for the study of ESD function, and also provides new insights into ESD in tumor suppression. Activation of ESD by the small-molecule chemical FPD5 could reduce the interaction between p53 and JAB1 and increased the protein level of p53 in the nucleus. p53 inhibited the expression of *CDCA8* and *CDC20*, arrested the cell cycle of A549 cells at the G0/G1 phase, and inhibited tumor growth.

## 5. Conclusions

In conclusion, this study revealed that ESD increased the protein level of p53 in the nucleus by JAB1 to inhibit the cell cycle of A549 lung cancer cells. p53 suppressed the gene expression of *CDCA8* and *CDC20* and arrested the cell cycle of A549 cells at the G0/G1 phase. Our findings are significant because the small molecule FPD5 activates the ESD enzyme activity to inhibit the nuclear export of p53, thereby inhibiting the growth of lung cancer, opening up a new approach for future studies of this novel ESD-mediated signaling pathway (Figure 7).

## Figures and Tables

**Figure 1 genes-13-00786-f001:**
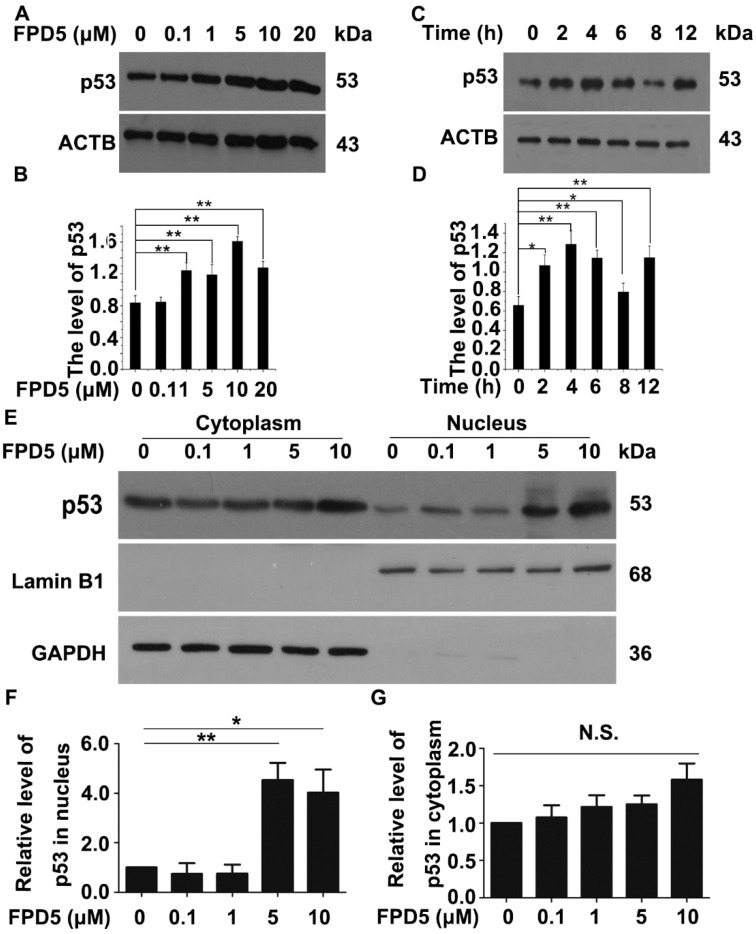
FPD5 increased the protein level of p53 in the nucleus of A549 cells. (**A**,**B**) The influence of different concentrations of FPD5 (0, 0.1, 1, 5, 10, 20 μM) for 24 h on the protein level of p53 in the A549 cell line was analyzed by Western blot. (**C**,**D**) A549 cells were treated with 5 µM FPD5 for 2, 4, 6, 8, and 12 h. The protein level of p53 was examined by Western blot. (**E**–**G**) The A549 cell line was treated with FPD5 (0, 0.1, 1, 5, 10 μM) for 24 h. The nucleus and cytoplasm were extracted for Western blot, respectively. Data are mean  ±  SEM. * *p* < 0.05, ** *p* < 0.01, N.S., not significant, *n* = 3.

**Figure 2 genes-13-00786-f002:**
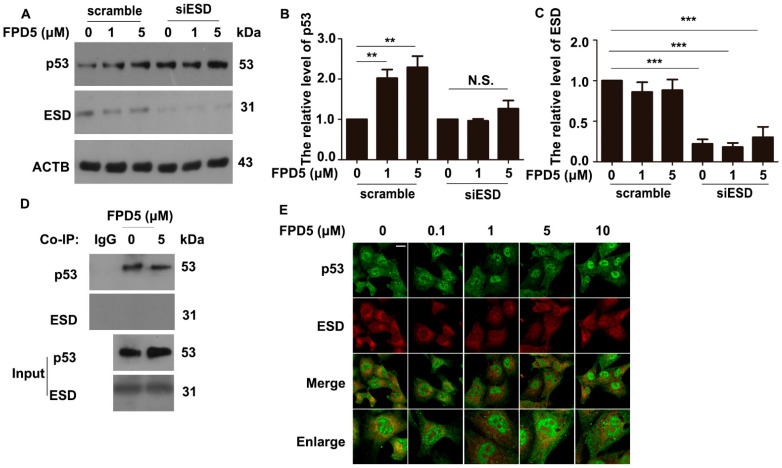
Activating ESD upregulated the p53 protein level in nucleus. (**A**–**C**) Western blot analysis of the protein level of p53 in A549 cells treated with 0, 1, and 5 µM FPD5 for 24 h after transfection with scramble siRNA or siRNA of ESD (siESD). (**D**) Co-immunoprecipitation (Co-IP) of ESD with the p53 protein in A549 cells. (**E**) Double immunofluorescence staining of p53 (green) and ESD (red) showed that there was no agglomeration or co-localization in A549 cells. Overlay of the p53 and ESD indicated that there was no interaction between them. Scale bar: 20 µm. Data are mean  ±  SEM. ** *p* < 0.01, *** *p* < 0.001, N.S., not significant, *n* = 3.

**Figure 3 genes-13-00786-f003:**
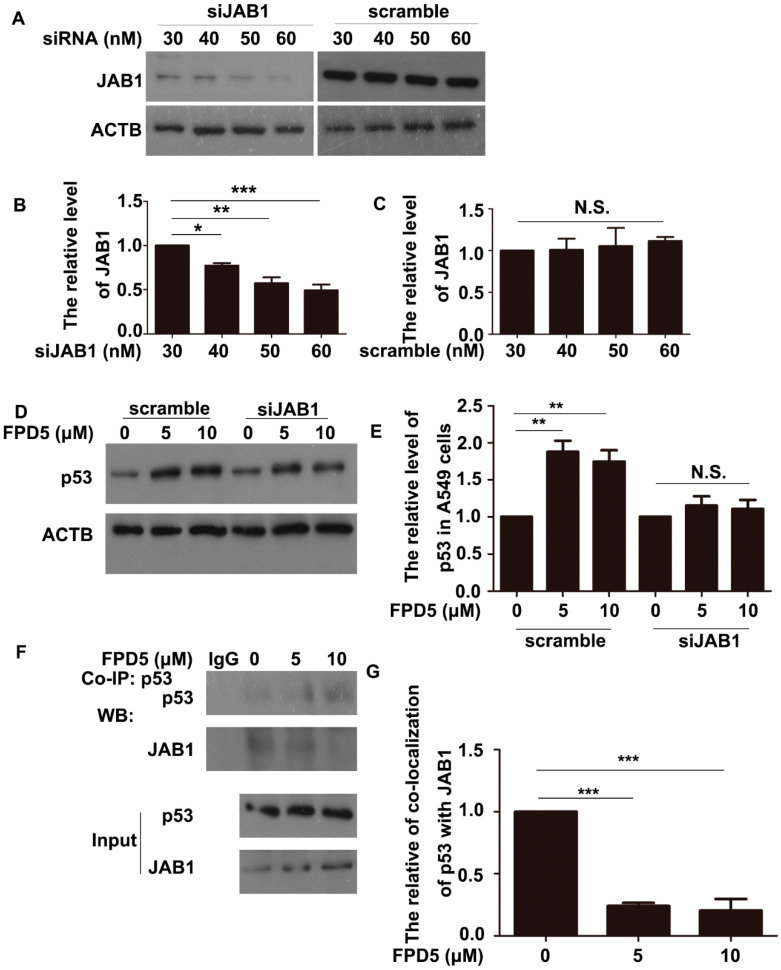
ESD improved p53 level by JAB1 and inhibited the interaction between JAB1 and p53. (**A**–**C**) Cells were transfected with scramble siRNA (scramble) or siRNA-JAB1 (siJAB1) for 24 h. The protein level of JAB1 was analyzed by Western blot, and the relative protein level of JAB1 was calculated as the ratio of JAB1 to ACTB. (**D**,**E**) Western blot analysis of p53 in A549 cells transfected with 60 nM scramble siRNA (scramble) or siRNA-JAB1 (siJAB1) for 24 h before exposure to 5, 10 μM FPD5 for 24 h. Relative protein level of p53 calculated as ratio of p53 to ACTB. (**F**,**G**) Co-immunoprecipitation (Co-IP) of JAB1 with the p53 protein in A549 cells. Data are mean  ± SEM. * *p* < 0.05, ** *p* < 0.01, *** *p*< 0.001, N.S., not significant, *n* = 3.

**Figure 4 genes-13-00786-f004:**
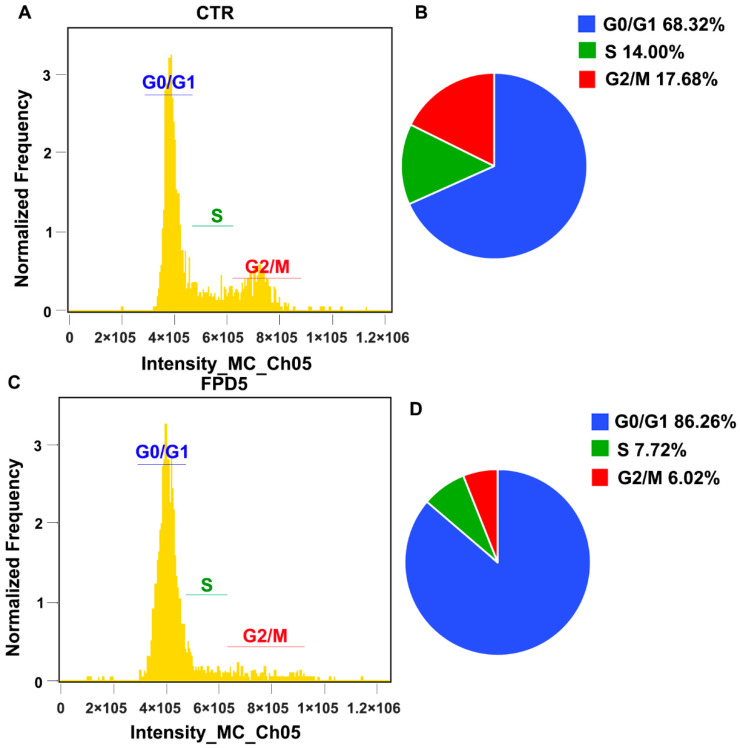
FPD5 arrested the cell cycle of A549 cells at G0/G1 phase. (**A**–**D**) After treatment with 5 μM FPD5 for 24 h, A549 cells were collected and stained with PI, and flow cytometry was performed to analyze the changes in the proportion of cells in each stage of the cell cycle. *n* = 3.

**Figure 5 genes-13-00786-f005:**
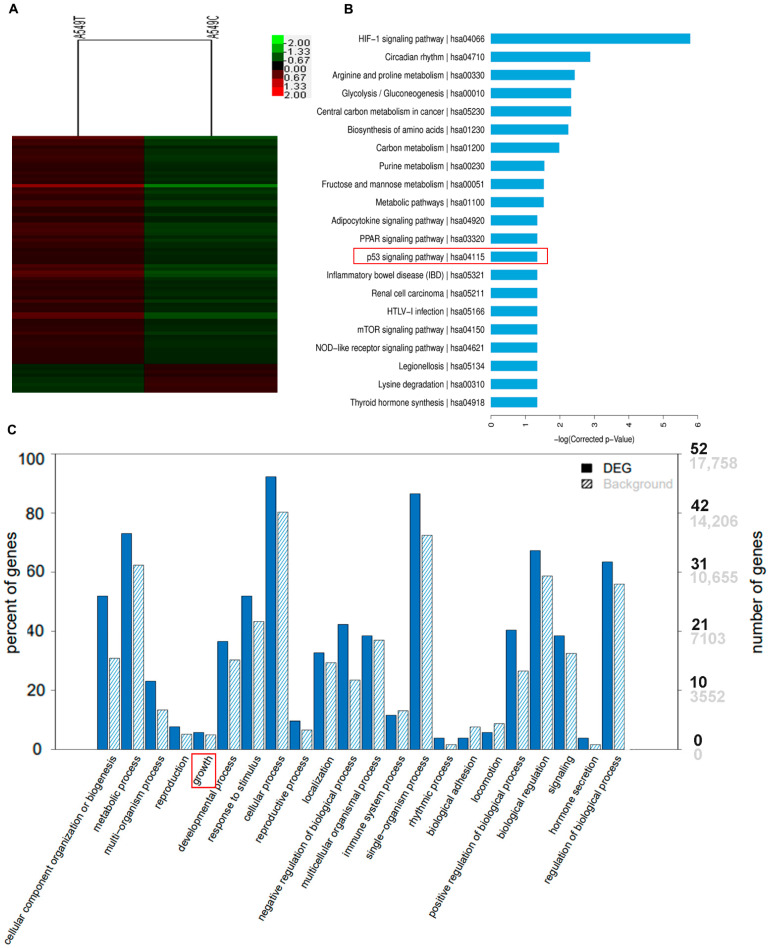
FPD5-treated A549 cell line for PrimeView Human Gene Expression Array. (**A**) Heatmap of differentially expressed genes in A549 cell line. (**B**) KEGG pathway enrichment analysis in A549 cell line. The red frame indicated the genes involve in the p53 signaling pathway. (**C**) GO enrichment classification map of genes regulated by ESD. The red frame indicated the genes involve in cell growth.

**Figure 6 genes-13-00786-f006:**
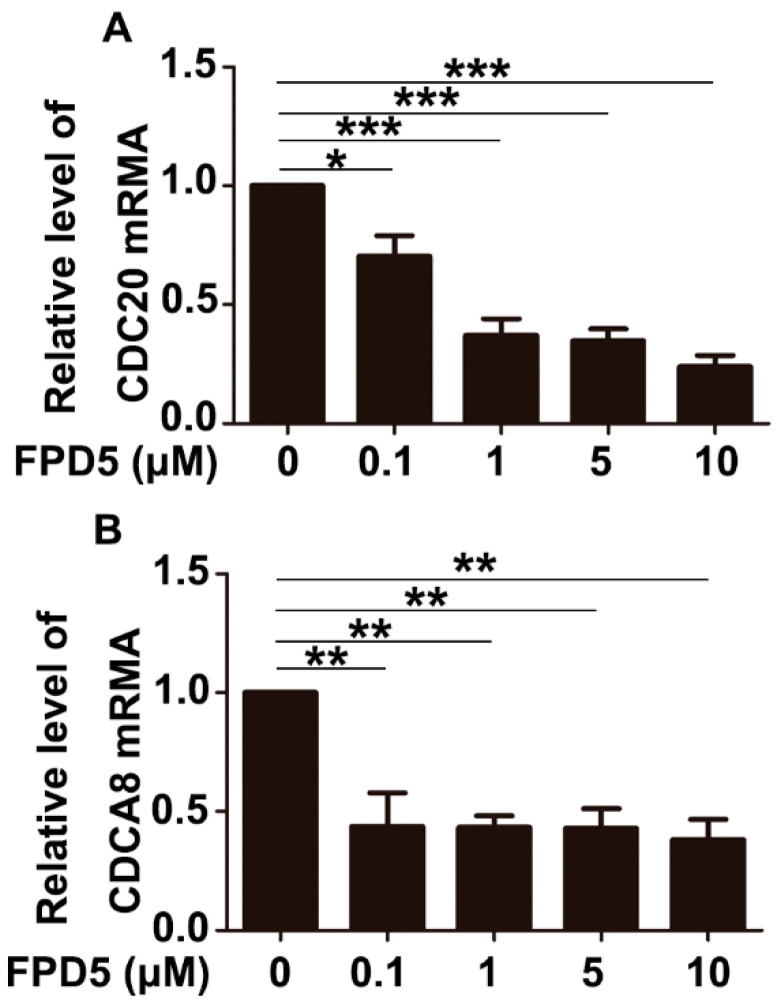
qPCR validation of p53 downregulated genes. (**A**,**B**): The mRNA levels of *CDCA8* and *CDC20* were detected by qPCR technology in A549 cells after treatment with 0, 0.1, 1, 5, 10 μM FPD5 for 24 h. Data are mean  ±  SEM. * *p* < 0.05, ** *p* < 0.01, *** *p* < 0.001, *n* = 3.

**Figure 7 genes-13-00786-f007:**
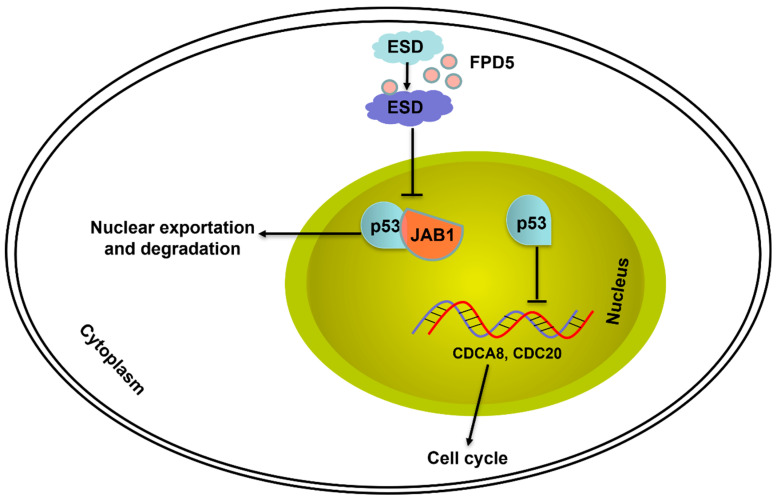
The mechanism of ESD inhibiting growth of A549 lung cancer cells. Activated ESD by FPD5 inhibited p53 to interact with JAB1, thereby reducing p53 exportation from nucleus to cytoplasm. p53 inhibited the expression of *CDCA8* and *CDC20* and thus arrested the cell cycle of A549 cells at G0/G1 phase and suppressed cell growth.

**Table 1 genes-13-00786-t001:** Fold change in genes downregulated by p53.

Gene Title	Gene Symbol	Log2 Ratio
Downregulated gene		
*Cell division cycle associated 8*	*CDCA8*	−1.50
*Cell division cycle 20*	*CDC20*	−1.74

## Data Availability

Not applicable.

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
