# Peer review of "Activation of Esterase D by FPD5 Inhibits Growth of A549 Lung Cancer Cells via JAB1/p53 Pathway"

_genes, 2022, doi:10.3390/genes13050786_

Round 1

Reviewer 1 Report

In this paper, Yao et al. investigated the interactions of FPD5 with p53, ESD and JAB1 via western blotting, immunoprecipitation and immunofluorescence. In addition, cell cycle analysis and qPCR were also carried out to study the impact of ESD and p53 on cellular proliferation. The paper clearly elucidates the molecular mechanism of the tumor suppressing effect of FPD5. The major findings of the paper are summarized as follows,

1) FPD5 elevates p53 in A549 at the whole-cell level.

2) FPD5 increases the nuclear protein level of p53.

3) The upregulation of p53 by FPD5 is moderated by ESD.

4) ESD regulates p53 via JAB1.

5) ESD reduces JAB1 binding to p53.

6) FPD5 inhibits cell cycle progression.

7) FPD5 suppresses CDCA8 and CDC20.

While most findings above are supported by the data, some claims need to be backed up with more rigorous experimental efforts and statistical analyses.

  1. The work was only performed in A549 cells (perhaps some tests were also done in HEK293T cells, which need to be clarified). It is important to confirm some results in at least another cell line to test the validity of the findings.
  2. As different concentrations of FPD5 is used, it is important to state whether proper vehicle control (e.g. DMSO, ethanol, etc.) was used to rule out any growth inhibitory effect due to differential vehicle concentrations.
  3. There are no lines/brackets indicating statistical comparisons. Therefore, it is confusing which post-hoc multiple t tests are statistically significant.
  4. When reporting fold changes, experimental errors should be shown for the baseline. For example, SEM is missing for 0 uM FPD5 in Fig 1F and 1G. Fig 3 B, C, E, G and Fig 6 A-B also need this correction.
  5. The authors claimed that ‘FPD5 significantly increased the overall 167 protein level of p53 in the cells in a concentration-dependent (Fig. 1A, 1B) and time-dependent (Fig. 1C, 1D) manner.’ However, it appears that there are only three different levels in Fig 1B: low: 0, 0.1; medium: 1, 5, 20; high: 10. The pattern also looks strange for Fig 1D. To be specific, the dose- and temporal-dependent behavior is neither monotonic increasing or leveling off. Rather, there is an increase at 10 uM FPD5 and a decrease at 8 h. The authors should explain more about this data.
  6. The authors stated that ‘compared with the cells transfected with scramble RNA, the protein level of p53 could not be up-regulated by FPD5 in A549 cells transfected with ESD siRNA (Fig. 2A-2C).’ Although the level of p53 does not change in a statistically significant manner under siESD, the authors need to explain why there is a downward trend for increasing concentrations of FPD5.
  7. The statistical significance in Fig 2B for 1 uM of FPD5 looks suspicious. Based on the overlap of SEM error bars, the p value should not be less than 0.05. The authors should report exact t test results for this comparison to justify the label of *.
  8. To support the statement that ESD regulates p53 via JAB1. The authors need to show how ESD activity is increased by FPD5 for the work performed in Fig 3.
  9. The authors should test at least one more concentration of FPD5 in addition to 5 uM to validate the conclusions made in Fig 3D-G.
  10. The discussion section of the paper needs to be expanded. Currently, it is short and does not fully describe the significance and generalization of all reported results.

Minor:

  1. The caption for Fig 3 F-G should be Co-IP of Jab1 with p53, not ESD.
  2. ‘Chemical small molecule’ should be referred to as ‘small molecule chemical.’
  3. ‘What’s more’ should be replaced by ‘In addition.’

Author Response

Point 1: The work was only performed in A549 cells (perhaps some tests were also done in HEK293T cells, which need to be clarified). It is important to confirm some results in at least another cell line to test the validity of the findings.

Response 1: Thanks for your useful advices. In this article, we mainly investigated the inhibitory effect of ESD on the growth of A549 cells. There were no tests done in HEK293T cell line and we had removed the text related to HEK293T in the section of “Materials and methods”.

The reason why we only performed the tests in A549 cells was that our research mainly focused on p53 wild-type tumor cells, and many reports have shown that about 50% of clinical tumor patients have wild-type p53 (1).

Meanwhile, we had verified that activation of ESD by FPD5 inhibited the cell viability in A549 cells, HeLa cells and H322 cells (2). So we focused on one of the tumors and pay more attention to the inhibition mechanisms of FPD5 in A549.

Point 2:As different concentrations of FPD5 is used, it is important to state whether proper vehicle control (e.g. DMSO, ethanol, etc.) was used to rule out any growth inhibitory effect due to differential vehicle concentrations.

Response 2: We thank the Reviewer for raising this concern. The proper control is important for getting convincing results and we did it in our experiment. Small molecule FPD5 was dissolved in DMSO. In our article, all of control groups (0 μM) were treated with DMSO with using the same volume as the highest concentration the same concentration of maximum FPD5 as of the treatment group.

Point 3:There are no lines/brackets indicating statistical comparisons. Therefore, it is confusing which post-hoc multiple t tests are statistically significant.

Response 3: Thanks for your useful advices again. We have checked all the figures carefully. All of statistical was compared with control group, and we had added lines to indicate statistical comparisons.

Point 4:When reporting fold changes, experimental errors should be shown for the baseline. For example, SEM is missing for 0 uM FPD5 in Fig 1F and 1G. Fig 3 B, C, E, G and Fig 6 A-B also need this correction.

Response 4: Thank you for your suggestion. In Fig 1F and 1G. Fig 3 B, C, E, G and Fig 6 A-B, we used the “Relative level” to report exhibit fold changes. The relative level of “0 uM FPD5” was 1, so there was no SEM for 0 uM FPD5.

Point 5:The authors claimed that ‘FPD5 significantly increased the overall 167 protein level of p53 in the cells in a concentration-dependent (Fig. 1A, 1B) and time-dependent (Fig. 1C, 1D) manner.’ However, it appears that there are only three different levels in Fig 1B: low: 0, 0.1; medium: 1, 5, 20; high: 10. The pattern also looks strange for Fig 1D. To be specific, the dose- and temporal-dependent behavior is neither monotonic increasing or leveling off. Rather, there is an increase at 10 uM FPD5 and a decrease at 8 h. The authors should explain more about this data.

Response 5: We thank the Reviewer for raising this concern. We are very sorry that our expression was inappropriate, and we have revised our statement in the manuscript: “It was found that FPD5 significantly increased the overall protein level of p53 in the cells (Fig. 1A, 1B, 1C, 1D), indicating that FPD5 could regulate the protein level of p53.” (lines 199-200 in revised manuscript). We will take more tests to research the reason why there is an increase at 10 uM FPD5 and a decrease at 8 h.

Point 6:The authors stated that ‘compared with the cells transfected with scramble RNA, the protein level of p53 could not be up-regulated by FPD5 in A549 cells transfected with ESD siRNA (Fig. 2A-2C).’ Although the level of p53 does not change in a statistically significant manner under siESD, the authors need to explain why there is a downward trend for increasing concentrations of FPD5.

Response 6: Thanks for your suggestion. We are sorry for the confusion caused by our data. In order to avoid unnecessary confusion caused by the data in Figure 2A, 2B, 2C, we repeated this experiment. We have re-stated the data and changed the Figure 2A, 2B, 2C in our revised manuscript.

Point 7:The statistical significance in Fig 2B for 1 uM of FPD5 looks suspicious. Based on the overlap of SEM error bars, the p value should not be less than 0.05. The authors should report exact t test results for this comparison to justify the label of *.

Response 7: Thanks for your suggestion. We are sorry for the confusion caused by our data. In order to avoid unnecessary confusion caused by the data in Figure 2A, 2B, 2C, we repeated this experiment. We have re-stated the data and changed the Figure 2A, 2B, 2C in our revised manuscript. The data was analyzed by One-Way ANOVA and the results are presented as follows:

Point 8:To support the statement that ESD regulates p53 via JAB1. The authors need to show how ESD activity is increased by FPD5 for the work performed in Fig 3.

Response 8: We thank the Reviewer for raising this concern. We had reported previously that FPD5 increased the activity of ESD (3).We had published ESD was activated by FPD5 and we had cited in references 11. Furthermore, we found that the protein level of p53 was upregulated in the A549 cells treated with 5 and 10 μM FPD5 after transfected with JAB1 siRNA. But FPD5 could no longer increase the overall protein level of p53 after knocking down the expression of JAB1 (Fig. 3D, 3E). Therefore, we concluded that ESD regulated p53 via JAB1.

For the reason, we have explained in the discussion. In our previous research, we found JAB1, an ESD-interacting protein, was acetylated on the 89th threonine residue. After activated by FPD5, ESD promoted its interaction with JAB1 and acetylated JAB1 at Thr89, resulting in enhanced the level of phosphorylation at this site (4).

Point 9:The authors should test at least one more concentration of FPD5 in addition to 5 uM to validate the conclusions made in Fig 3D-G.

Response 9: Thanks for your suggestion. As suggested, we supplemented the experiment. The A549 cells were treated with 5 and 10 μM FPD5 after transferred with 60 nM siJAB1 and the level of p53 was not up-regulated by FPD5 (Fig.3D, 3E). In addition, the interaction between p53 and JAB1 was significantly suppressed by 5 and 10 μM FPD5 (Fig.3F, 3G).

Point 10:The discussion section of the paper needs to be expanded. Currently, it is short and does not fully describe the significance and generalization of all reported results.

Response 10: Thanks for your suggestion again. In the discussion, we discussed the possibility of ESD inhibiting the interaction between JAB1 and p53 and the significance of our study in Lines 308-350 in our reversed manuscript:

ESD is a non-specific esterase widely distributed in mammalian cells. Today, more and more studies have focused on the relationship between ESD activity and multiple pathological processes such as retinoblastoma, lung adenocarcinoma, viral immunity, etc (7, 25-31). However, there are few studies on ESD in tumor suppression. The direct regulation of specific target proteins and signaling pathways by small molecule drugs is widely used in cancer therapy (32). In this study, we found for the first time that ESD inhibited the cell cycle progress of A549 cells through JAB1 / p53 signal pathway and had a significant inhibitory effect on tumor growth. This will not only provide new therapeutic targets and schemes for the clinical treatment of lung cancer, but also provide a theoretical basis for the development of new drugs to inhibit tumor growth.

P53 regulates the expression of genes involved in regulating cell proliferation and growth in the nucleus (33). There is an increasing evidence that mutated p53 plays an important role in tumorigenesis, invasion and migration (34). JAB1, also names as the 5th subunit of COP9 signalosome (CSN5 or COPS5), interacts with Hdm2 to promote p53 nuclear exportation. Then MDM2 mediated p53 ubiquitination and degradation in cytoplasm (35, 36). In this study, we found that activating ESD by FPD5 significantly inhibited the interaction between JAB1 with p53, which would reduce the effect of nuclear exportation of p53 by JAB1, which increased the nuclear localization of p53 and provided a good condition for p53 to regulate other gene expression in the nucleus. However, we speculated that there are two possibilities for the reason why activated ESD inhibited the interaction between JAB1with p53. The first possibility is that JAB1 can no longer bind to p53 after interacting with ESD. And the second possibility is that the acetylation modification of JAB at Thr89 is very important for its binding to p53. The acetylation modification of JAB1 at thr89 was removed by ESD after activated by FPD5(37), resulting in the reduction of the binding ability to p53. We will research the effect of this site on the binding ability of JAB1 to p53 through the threonine point mutant T89A for further verification.

In summary, our study discovered for the first time the molecular mechanism of ESD inhibiting tumor growth through the JAB1/p53 signaling pathway, which will provide a good theoretical basis for the study of ESD functional diversity, and also provide new insights into ESD in tumor suppression. Activation of ESD by the small molecule chemical FPD5 could reduce the interaction between p53 with JAB1 and increase the protein level of p53 in the nucleus. P53 inhibited the expression of CDCA8 and CDC20 and arrested the cell cycle of A549 cells at G0/G1 phase and inhibited tumor growth.

Minor:

Point 1:The caption for Fig 3 F-G should be Co-IP of Jab1 with p53, not ESD.

Response 1: Thanks for your useful advices again. We are very sorry for we mistakenly wrote JAB1 as ESD in the Figure legend of Fig 3F-G. We revised the Figure legend of Fig 3F-G as ‘(F, G) Co-immunoprecipitation (Co-IP) of ESD JAB1 with the p53 protein in A549 cells’ in Lines 266-267 in our reversed manuscript.

Point 2:‘Chemical small molecule’ should be referred to as ‘small molecule chemical.’

Response 2: Thanks for your useful advices again. We revised ‘Chemical small molecule’ as ‘small molecule chemical’ in our reversed manuscript.

Point 3:‘What’s more’ should be replaced by ‘In addition.’

Response 3: Thanks for your useful advices again. We revised ‘What’s more’ as ‘ In addition’ in Line 244 in the reversed manuscript.

Reviewer 2 Report

Authors presented interesting study elucidating the mechanism of esterase D activation by FPD5 on the nuclear p53 level. The changes of p53 concentration affect the cell cycle (G0/G1 arrest) and cell-cycle-related gene expression (CDCA8 and CDC20). Obtained data suggest the presence of functional link between JAB1 modifcation by esterase D (putatively previously described deacetylation at Thr 89) and increased concentration of p53 in nucleus. Research is generally well planned and performed, obtained results suport presented hypotheses.  Some minor corrections are necessary, before the article could be published,  mainly in relations to more precise description of material and methods section. Details are presented below.

Line 73, 173, 216,233  – lack of space before [23], [13], (Fig. 3D, 3E), and [25] respectively

Line 97- immunoprecipitated complexes attached to agarose beads should be isolated by centrifugation , for example 3000g for 2-3 min, and then washed at least three times using washing buffer. Provide details of centrifugation (g force and time), volume, composition and number of washing repetitions using the washing buffer.

Line 101- the number of kit 78833 should be removed, instead provide a full name of kit manufacturer.  

Section 2.8

Amount of RNA taken for analysis, how the quality of RNA was assessed- for example using Nanodrop or related equipment, details of PCR reaction, citation of Livak and Schmittgen method.

Line 133-135; provide the 5’-3’orientation for tested gene primers

Fig 1 F,G and 3G; no space FPD5(µM)

Line 232- in vivo, in vitro should be in italics

References

The reference nr 23 and 24 are the same, remove one of them.

Generally is provided name of only the first Author, more Author names should be presented; if more that 6 Authors are present, then write et  al.

Author Response

Dear Editor in Chief,

Thank you very much for your and reviewers' appropriate and constructive comments concerning our paper (genes-1666005). We took all criticisms into great consideration and revised the manuscript accordingly. Changes are shown in red in the text and all the figures in the manuscript are added at the end. We believe we appropriately addressed the raised concerns and improved rigor and quality of the work. We hope you and the Reviewers are satisfied with our revision work and recognize our effort. Our itemized responses to each comment follow.

Reviewer reports:

Review report 2:

Authors presented interesting study elucidating the mechanism of esterase D activation by FPD5 on the nuclear p53 level. The changes of p53 concentration affect the cell cycle (G0/G1 arrest) and cell-cycle-related gene expression (CDCA8 and CDC20). Obtained data suggest the presence of functional link between JAB1 modifcation by esterase D (putatively previously described deacetylation at Thr 89) and increased concentration of p53 in nucleus. Research is generally well planned and performed, obtained results suport presented hypotheses.  Some minor corrections are necessary, before the article could be published, mainly in relations to more precise description of material and methods section. Details are presented below.

  1. Line 73, 173, 216,233 – lack of space before [23], [13], (Fig. 3D, 3E), and [25] respectively

Response: We are very grateful to the reviewer’s comments. Because the references 23 and 24 are the same in our previous manuscript, we removed the references 23 in Line 127 in the revised manuscript. And we supplemented the space before [13], (Fig. 3D, 3E), and [25] in Line 203, 249 and 267 in the revised manuscript, respectively.

  1. Line 97- immunoprecipitated complexes attached to agarose beads should be isolated by centrifugation , for example 3000g for 2-3 min, and then washed at least three times using washing buffer. Provide details of centrifugation (g force and time), volume, composition and number of washing repetitions using the washing buffer.

Response: We are very grateful to the reviewer’s comments. We supplemented the experimental steps of co-immunoprecipitation in Lines 99-105 in the revised manuscript. The new step is ‘For co-immunoprecipitation, lysates (0.5 mg of protein) of cells were incubated with 1.0 μg of primary antibody (2.0 μg) overnight for 24 h at 4 °C, containing with 20 μL of protein A/G Sepharose beads (Beyotime). The immunoprecipitated complexes at-tached to agarose beads were washed three times with 400 μL PBS (containing 1 mM PMSF) and then isolated by centrifugation at 4 ℃, 2000 rpm for 5 min. Then the immunoprecipitates were eluted with IP lysate and boiled with 25 μL 2 × SDS loading buffer at 100 ℃ for Western blot.’

  1. Line 101- the number of kit 78833 should be removed, instead provide a full name of kit manufacturer.

Response: We are very grateful to the reviewer’s comments. The full name of kit is ‘NE-PERTM Nuclear and Cytoplasmic Extraction Reagents kit (ThermoFisher Scientific, 78835)’. We supplemented this full name to Lines 109-110 in the revised manuscript.

Section 2.8

  1. Amount of RNA taken for analysis, how the quality of RNA was assessed- for example using Nanodrop or related equipment, details of PCR reaction, citation of Livak and Schmittgen method.

Response: We are very grateful to the reviewer’s comments. The quality of RNA was assessed by Nanodrop 2000 equipment (ThermoFisher Scientific). We revised it in Line 150-151 in the revised manuscript.

The action was cycled for 40 times including Pre denaturation at 95 ℃ for 5 min, de-naturation at 95 ℃ for 30 s, annealing at 58 ℃ for 30 s and extension at 72 ℃ for 30 s. We revised it in Line 165-166 in the revised manuscript.

  1. Line 133-135; provide the 5’-3’orientation for tested gene primers

Response: We are very grateful to the reviewer’s comments. As suggested, we provided the 5’-3’orientation for CDCA8 and CDC20 gene primers in Line 154-157 in the revised manuscript.

  1. Fig 1 F, G and 3G; no space FPD5(µM)

Response: We are very grateful to the reviewer’s comments. As suggested, we supplemented the space and revised in      Fig 1 F, G and 3G in the revised manuscript.

  1. Line 232- in vivo, in vitro should be in italics

Response: We are very grateful to the reviewer’s comments. As suggested, we changed the font of ‘in vivo, in vitro’ to Italic in Line 268 in the revised manuscript.

References

  1. The reference nr 23 and 24 are the same, remove one of them.

Response: We are very grateful to the reviewer’s comments. As suggested, we removed the reference nr 23 in Line 127 in the revised manuscript.

  1. Generally is provided name of only the first Author, more Author names should be presented; if more than 6 Authors are present, then write et al.

Response: We are very grateful to the reviewer’s comments. As suggested, we change the format of the section of ‘References’.

Figure 1. FPD5 increased the protein level of p53 in the nucleus of A549 cells. (A, B) The influence of different concentrations of FPD5 (0,0.1,1,5,10,20 μM) for 24 h on the protein level of p53 in A549 cell line was analyzed by Western blot. (C, D) A549 cells were treated with 5 µM FPD5 for 2, 4, 6, 8 and 12 h. The protein level of p53 was examined by Western blot. (E, F, G) A549 cell line was treated with FPD5 (0, 0.1, 1, 5, 10 μM) for 24 h. The nucleus and cytoplasm were extraction for Western blot respectively. Data are mean ± SEM. *p < 0.05,**p < 0.01,***p < 0.001, n=3.

Figure 2. Activating ESD up-regulated the p53 protein level in nuclear. (A, B, C) Western blot analysis of the protein level of p53 in A549 cells treated with 0, 1 and 5 µM FPD5 for 24 h after transfected with scramble siRNA or siRNA of ESD (siESD). (D) Co-immunoprecipitation (Co-IP) of ESD with the p53 protein in A549 cells. (E) Double immunofluorescence staining of p53 (green) and ESD (red) showed there was no agglomeration and co-localization in A549 cells. Overlay of the p53 and ESD indicated there was no interaction between them. Scale bar: 20 µm. Data are mean ± SEM. *p < 0.05, ***p < 0.001, n=3.

Figure 3. ESD improved p53 level by JAB1 and inhibited the interaction between JAB1 with p53. (A, B, C) were transfected with scramble siRNA (scramble) or siRNA-JAB1 (siJAB1) for 24 h. The protein level of JAB1 was analyzed by Western blot, and relative protein level of JAB1 was a ratio of JAB1 to ACTB. (D, E) Western blot analysis of p53 in A549 cells transfected with 60 nM scramble siRNA (scramble) or siRNA-JAB1 (siJAB1) for 24 h before exposed 5,10 μM FPD5 for 24 h. Relative protein level of p53 is a ratio of p53 to ACTB. (F, G) Co-immunoprecipitation (Co-IP) of JAB1 with the p53 protein in A549 cells. Data are mean ± SEM. *p < 0.05, **p < 0.01, ***p< 0.001, n=3.

Figure 4. FPD5 arrested the cell cycle of A549 cells at G0/G1 phase. (A-D) After treated with 5 μM FPD5 for 24 h, A549 cells were collected and stained with PI, and flow cytometry was performed to analyze the changes in the proportion of cells in each stage of the cell cycle. n=3.

Figure 5. FPD5 treated A549 cell line for PrimeView Human Gene Expression Array. (A,) Heatmap of differentially expressed genes in A549 cell line. (B) KEGG pathway enrichment analysis in A549 cell line. (C) GO enrichment classification map of genes regulated by ESD.

Figure 6. qPCR validation of p53 downregulated genes. (A-C):The mRNA levels of CDCA8 and CDC20 were detected by qPCR technology in A549 cells after treated by 0, 0.1, 1, 5, 10 μM FPD5 for 24 h. Data are mean ± SEM. *p < 0.05,**p < 0.01,***p < 0.001, n=3.

Figure 7. The mechanism of ESD inhibiting growth of A549 lung cancer cells. Activated ESD by FPD5 inhibited p53 to interact with JAB1, thereby reduced p53 exportation from nucleus to cytoplasm. P53 inhibited the expression of CDCA8 and CDC20 so that arrested the cell cycle of A549 cells at G0/G1 phase and suppressed cell growth.

Round 2

Reviewer 1 Report

The majority of my comments have been addressed satisfactorily.

Regarding Point 4, all baseline conditions should be presented with error bars. As each condition contains replicates, SEM can always be calculated. Please refer to 10.1016/j.bcp.2007.11.021 as an example.

As a quick demo, consider base line conditions (1, 1.2, 1.3) and comparison group (1.8, 2, 2.2), after fold change calculations (for relative level), the data becomes:

Baseline: (0.857, 1.029, 1.114)

Comparison group: (1.543, 1.714, 1.886)

The stats are therefore as follows:

Baseline: Average: 1.00; SEM: 0.076

Comparison group: Average: 1.71; SEM: 0.099

The SEM of 0.076 should be included for the baseline condition.

Author Response

Response to Reviewer Comments

Dear Editor in Chief,

Thank you very much for your and reviewers' appropriate and constructive comments concerning our paper (genes-1666005). We took all criticisms into great consideration and revised the manuscript accordingly. Changes are shown in red in the text. We believe we appropriately addressed the raised concerns and improved rigor and quality of the work. We hope you and the Reviewers are satisfied with our revision work and recognize our effort. Our itemized responses to each comment follow.

Reviewer reports:

Point 1:Regarding Point 4, all baseline conditions should be presented with error bars. As each condition contains replicates, SEM can always be calculated. Please refer to 10.1016/j.bcp.2007.11.021 as an example.

As a quick demo, consider base line conditions (1, 1.2, 1.3) and comparison group (1.8, 2, 2.2), after fold change calculations (for relative level), the data becomes:

Baseline: (0.857, 1.029, 1.114)

Comparison group: (1.543, 1.714, 1.886)

The stats are therefore as follows:

Baseline: Average: 1.00; SEM: 0.076

Comparison group: Average: 1.71; SEM: 0.099

The SEM of 0.076 should be included for the baseline condition.

Response 1:We thank the Reviewer for raising this concern. In Fig 1F and 1G. Fig 3 B, C, E, G and Fig 6 A-B, we used the “Relative level” to report exhibit fold changes. It means that all groups, including control group and comparison group, are divided by the baseline. The ‘Relative’ level of control group was 1 in 3 independent replicates, so there were no error bars for baseline.

For example:

The level:

Independent replicates

Group

First

Second

Third

Baseline

0.857

1.029

1.114

Comparison group

1.543

1.714

1.886

The Relative level (All groups were divided by the baseline):

Independent replicates

Group

First

Second

Third

Baseline

0.857/0.857=1

1.029/1.029=1

1.114/1.114=1

Comparison group

1.543/0.857=1.800

1.714/1.029=1.666

1.886/1.114=1.693
